# Impact on Education and Ecological Footprint as a Consequence of SARS-CoV-2 in the Perception of the Quality of Teaching Engineering Students in the Brazilian Amazon

**Luiz Maurício Furtado Maués** [1,*] **, Felipe de Sá Moreira** [2] **, Luciana de Nazaré Pinheiro Cordeiro** [1]**,
Raísse Layane de Paula Saraiva** [1] **, Paulo Cerqueira dos Santos Junior** [3] **and Olga Maria Pinheiro Pinheiro** [1]

[1]  Programa de Pós-Graduação em Engenharia Civil, Instituto de Tecnologia, Campus Universitário Guamá,
    Universidade Federal do Pará (UFPA), Belém 66075-110, PA, Brazil
[2]  Faculdade de Engenharia Civil, Instituto de Tecnologia, Campus Universitário Guamá,
    Universidade Federal do Pará (UFPA), Belém 66075-110, PA, Brazil
[3]  Programa de Pós-Graduação em Matemática e Estatística, Instituto de Ciências Naturais,
    Campus Universitário Guamá, Universidade Federal do Pará (UFPA), Belém 66075-110, PA, Brazil
*   Correspondence: maues@ufpa.br

**Abstract:** The world experienced several economic, social and environmental transformations during the COVID-19 pandemic, and today, society assesses all these changes in the different stages of the pandemic process. In this sense, this research aimed to evaluate the educational and environmental impacts on the academic community of the largest educational institution in the Brazilian Amazon. The perception of the quality of teaching of undergraduate and graduate engineering students during remote teaching was assessed by means of an exploratory research work carried out at the Federal University of Pará (Brazil). Ecological and carbon footprint indicators were also measured through the Global Footprint Network®. In social terms, students pointed to a reduction in the quality of teaching. Despite the incentives through institutional initiatives for both students and professors, the community was not prepared to fully migrate to the virtual world, and that made the teaching–learning process difficult. In environmental terms, the reduction in the ecological footprint that was observed could have reached values higher than those that were found, to the order of 2.33%, with the mobility sector achieving the greatest reduction.

**Keywords:** COVID-19 pandemic; education; ecological footprint

## 1. Introduction

The impact that the severe acute respiratory syndrome (SARS-CoV-2) responsible for the coronavirus disease (COVID-19) has had on the planet since it started spreading is notorious. It began in late December 2019, when the first case was recorded in China in the province of Hubei, in the capital Wuhan [1]. It quickly spread across Europe in February 2020 [2] and other continents, which prompted the United Nations (UN) to declare a pandemic. The main and most damaging consequence was the loss of hundreds of thousands of lives as a result of the rapid proliferation of the virus and evolution of patients' condition. In Brazil, the first case was registered in the city of São Paulo on 26 February 2020. Since then, problems of various kinds have been increasing, thus forcing governments to take drastic measures, resulting in changes in the usual behavior of society, whether economic, social or environmental.

Regarding the economy, countries implemented a reduction in their activities, which generated a change in conduct in the management of businesses and social problems, as found in the research by Bartik et al. [3], which states that in the United States of America around 30 million jobs were displaced due to the pandemic, which is the highest rate since the Great Depression.

In the social area, restrictions that were imposed on the population such as going to work, shopping, traveling, leisure, among others, were widely reported by governments in several countries [4]. These actions were intended to reduce the spread of the virus [5] among the population, but they also more significantly affected the low-income population [6] and populations with disabilities [7].

The COVID-19 pandemic also generated restrictions in the field of education due to the prohibition of face-to-face classes through national decrees; consequently, institutions adopted online classes as an alternative so that students could continue studying. This sudden change in pedagogical processes forced educators, students and the Brazilian academic community in general to undergo adaptations in their educational routines.

The adaptations the academic community experienced can be seen in a survey carried out by McFadden et al. [8], who sought to assess the impact of the pandemic on universities in Australia, England, Finland, Northern Ireland, Norway, Ireland and Sweden. It was found that there were changes both in the admission phase of these institutions and in the implementation of virtual classes for students of the Social Sciences course. This change in behavior was recurrent in universities in both developed and developing countries.

At the Universidade Federal do Pará (UFPA) [Federal University of Pará], face-to-face activities were called off on 19 March 2020 and work groups were set up to plan for the resuming of activities, even if in a virtual format. The institution has 11 campuses throughout Pará State; it is the higher education institution with the most campuses in a state in Brazil, and it offers teaching, research and extension courses that provide good quality education to people in rural areas. Due to the institutional scope, some planning was required to resume classes virtually. In July 2020, courses and workshops on the digital tools available and on the strategies that could be adopted throughout the teaching–learning processes began to be offered to professors. Additionally, on 21 August 2020, through Resolution No. 5.294, Emergency Remote Teaching was exceptionally and temporarily approved at different levels of education for the courses offered by the University due to the pandemic situation prompted by the new Coronavirus-COVID-19.

Civil Engineering stands out as the core of engineering in Brazil. In Pará, the first course was founded in 1931, and due to the breadth of the curriculum new aspects of this area were originated. For example, the Engenharia Sanitária e Ambiental (ESA) (Sanitary and Environmental Engineering) course, was created in 2013 at UFPA. With an audience 3 times smaller than that of Civil Engineering, ESA has excelled in approaches regarding the harmonious coexistence between man and the environment versus a traditional approach, with the strong technological bias that is found in the Civil Engineering course.

Understanding how these students approach environmental issues is important for understanding the role of the engineer in sustainable development. After all, four decades have passed since emergence of one of the first world movements regarding the preservation of nature, the Stockholm Declaration. Additionally, the construction industry still stands out for promoting environmental impacts.

A limitation found in this research refers to the application of the questionnaire in only two engineering courses of this institution. Due to the small number of courses and students involved in the sample, the conclusions may not be generalized. However, in order to extend the study to a representative sample, a greater number of professors participating in this research would be required so that they could conduct the questionnaire with their students, and that could hardly occur due to the isolation prompted by the COVID-19 pandemic.

Despite worldwide efforts to develop a vaccine to immunize the population, the fact remains that this pandemic changed people's routines, mainly in terms of travel restrictions and change in work status and education, which were instead carried out remotely due to the risk of contagion and viral dissemination—this change in behavior has been called the "new normal".

In this context, questions arise about the impact of this new reality in relation to the quality of learning during this year of significant changes in the educational process, where

educational institutions needed to make adaptations in their operational mode and began to offer virtual classes.

On the other hand, there are positions that emphasize the benefit of this new reality, which was compulsorily adopted by governments for society, especially in the field of sustainability, where fewer natural resources have been consumed and there has been a reduction in the generation of Greenhouse Gases (GHG) on the planet [9]. Although identifying benefits to the environment is complex, the authors Lenzen et al. [10] carried out an analysis of GHG reduction and concluded that there was a significant reduction—the largest in the history of the planet—due to COVID-19. Several authors have reported this positive relation.

To exemplify some of these statements, the research by Bera et al. [11] found that in India the concentration levels of gases ($NO_2$ and aerosol) were significantly lower in the lock-down period, either due to the reduction in economic activities or the traffic of vehicles. In South Korea, there was also an improvement in air quality due to the pandemic [12] as well as in Argentina, where it was also possible to verify the improvement in air quality and meteorological variables due to the pandemic [13].

Despite the relevance of the topic, no studies were found in the literature that evaluate footprints in this scenario called the new normal. In this context, seeking to address this knowledge gap, this research aimed to evaluate the ecological footprint and its impacts during this pandemic period among graduate and undergraduate students in the largest educational institution in the Amazon region. This article aimed to answer the following research questions:

1.  What is the student's perception regarding the quality of the classes taught and the level of learning by adopting virtual classes?
2.  How large is the ecological footprint of these students when attending face-to-face classes?
3.  What was the reduction in the students' ecological footprint when the courses adopted virtual classes?

In order to support the study, three theoretical reference topics were developed, which were divided into Ecological Footprint; tools for its calculation and the teaching–learning process in the pandemic, which are presented below.

## 2. Literature Review

### 2.1. Ecological Footprint

According to Tian et al. [14], the planet's population increased by 200 million people between 1900 and 2018, reaching a significant total of 4.22 billion inhabitants. In addition to this population increase, another noteworthy phenomenon is the number of inhabitants in urban areas in relation to those who live in rural areas. According to the UN [15], the population living in urban areas accounts for over 56% of the total world population. In the state of Pará, the target of the research, according to the latest census by the Instituto Brasileiro de Geografia e Estatística (IBGE) (Brazilian Institute of Geography and Statistics) [16], 68.5% of the total population live in urban areas. This process of population migration from the countryside to the city that has been occurring over the years, has effectively contributed to a process of increasing pollution in urban land and groundwater resulting from anthropogenic actions [17,18].

One of the methodologies adopted to assess the environmental impact caused by products is the carbon footprint. According to ISO/TS 14067:2018 [19], the carbon footprint can be defined as a measure of climate change arising from a product as a function of the Greenhouse Gases (GHG) that are emitted during the product's life cycle. Since GHGs are mainly made up of carbon dioxide ($CO_2$), methane ($NH_4$), nitrous oxide ($N_2O$), among others, one of the most important is $CO_2$ [20]. According to Rossi and Sales [21], in order to obtain a homogeneous dimensional unit of measurement, the different types of gases are transformed into their carbon dioxide equivalent ($CO_2$-eq), i.e., in a first step, the GHG generated in the production process is calculated and then transformed into $CO_2$-eq [22].

For this purpose, in the 1990s, the concept of the footprint was developed and introduced as a method capable of measuring the consumption of natural resources and the unsustainability that man causes in the environment [23]. The methodology for calculating the footprint includes the consumption of energy, agricultural land, pastures, built-up land, sea and forest as references [24] and seeks to calculate whether or not there is an excess of consumption in relation to the biocapacity of the planet, expressed in the Global Hectare (GHA) unit per capita.

Footprint quantification has been applied in several segments in society. One example is the calculation of the footprint in the transportation area, where the large volume of vehicles that transport people to work, to school, to go shopping or for leisure becomes relevant. This fact was corroborated by Rashid et al. [24], as they stated that there is a direct correlation (R2 = 0.7446) between the distance traveled by vehicles and the increase in footprint. In another study, the footprint was also calculated to measure the reduction in pollutant emissions when the population uses the city's public transportation system more intensively [25].

Another important factor is energy consumption in cities, as this resource is responsible for a large part of the energy footprint in countries such as China [26]. Another causative factor that can be assessed through the ecological footprint refers to the types of construction, since buildings that use heavy materials in the production process (such as steel and cement) generate a greater amount of pollutant emissions [27,28].

Therefore, it was necessary to assess the impact caused by these changes in relation to our needs for natural resources and the planet's ability to regenerate due to the historically occurring changes on the planet. For this purpose, researchers use the Life Cycle Assessment (LCA) methodology, which is a tool applied in several areas, whether in the extraction of raw materials or in the production process of finished products [29]. However, due to the complexity of calculating the environmental impacts when using the LCA, research is developed using the footprint, as this methodology is easier to apply and to be understood by wider society [30,31].

## 2.2. Tools for Calculating the Ecological Footprint

The ecological footprint is a methodology that provides measurements of the demand that compete for biologically productive space through biocapacity and footprint [32], and several tools have been developed to meet this demand, such as the ecological footprints of carbon, water, materials, and the earth [33].

It is possible to mention some of the widespread applications of these tools, such as those that were developed with the aim of evaluating the impact generated in products and in organizations. A methodology developed by the European Commission in 2012 aims to calculate the footprint in organizations [34] in order to identify environmental issues in the operational process of companies. Tools that involve the concept of Building Information Modeling (BIM) have also been applied to calculate $CO_2$ emissions, for example in housing construction [35]. Pellegrini et al. [36] applied the Water Footprint tool to verify the impact generated by the consumption of water by agriculture. The authors Florindo et al. [37] applied the combined carbon footprint and LCA technique to gauge the impact generated by four different types of livestock production systems.

Nowadays, with the dissemination of sustainability concepts and the increasing importance of the impact of people's consumption on the environment, there has been a growing development of tools available on the web, thus allowing for the calculation of the size of the footprint. These tools have different characteristics, with their own advantages and limitations. To exemplify the wide variety of tools available for calculating the Ecological Footprint, some are listed in Table 1.

**Table 1.** Software for calculating the ecological footprint.

| Tools | Country of Origin | Access Type | Developer | URL * |
|---|---|---|---|---|
| Calculation Tools | U.S. | Free | Greenhouse Gas Protocol | https://ghgprotocol.org/calculation-tools |
| Carbon emissions calculation tool Highways England | U.K. | Free | Highways | https://nationalhighways.co.uk/suppliers/carbon-emissions-calculation-tool/ |
| Carbon Footprint | U.K. | Free/Paid out | Carbon Footprint Ltd. | https://advanced-uk.com/esg-co2-reporting-from-99-month/?gclid=EAIaIQobChMIpff-q47B-AIVAWuRCh08cwMNEAAYASAAEgJGcfD_BwE |
| Carbon Footprint Calculator | U.S. | Free | Environmental Protection Agency | https://www3.epa.gov/carbon-footprint-calculator/ |
| CarbonCloud | Sweden | Paid out | CarbonCloud | https://carboncloud.com/get-started/ |
| Construction carbon calculator | U.S. | Free | Build Carbon Neutral | http://www.buildcarbonneutral.org/ |
| eToolLCD | U.K. | Paid out | Etool | https://etoolglobal.com/ |
| Global Footprint Network | U.S. | Free | Global Footprint Network | https://www.footprintnetwork.org/resources/footprint-calculator/ |
| Hotel Carbon Measurement Initiative | Denmark | Paid out | Sustainable Hospitality Alliance | https://sustainablehospitalityalliance.org/resource/hotel-carbon-measurement-initiative/ |
| IMPACT Compliant Tools | U.S. | Paid out | BRE Group | https://bregroup.com/products/impact/impact-compliant-tools/ |
| One Click LCA | Finland | Paid out | About Bionova Ltd. | https://www.oneclicklca.com/bionova-becomes-one-click-lca/ |
| SINAi | U.S. | Paid out | SINAI Technologies | https://www.sinaitechnologies.com/ |
| SME Carbon Footprint Calculator | U.K. | Free | Carbon Trust | https://www.carbontrust.com/resources/sme-carbon-footprint-calculator |
| Umberto LCA + | Germany | Paid out | Ifu Hamburg | https://www.ifu.com/umberto/lca-software/ |

* Accessed on 10 February 2021.

Due to the various options it provides–and as it is not the objective of this research to evaluate the performance of the tools–the tool provided by Global Footprint Network® (GFN – Oakland, CA, USA) was adopted in this research. According to Franz, Papyrakis [38] and Collins et al. [39], this footprint calculator is one of the most comprehensive due to the information contained in the questionnaire. Besides, this tool is free and allows the user to implement it in the mother language of the target city of the research, which facilitates the understanding of the concept and a greater assertiveness of the answers.

*2.3. Teaching–Learning Process during the Pandemic*

The pandemic has promoted a change in educational systems based on the delivery of online content as a way of adapting the teaching–learning process. According to UN-ESCO [40], around 90% of students worldwide had their classroom activities interrupted. According to the Brazilian Institute of Geography and Statistics [41], one out of four people in Brazil do not have access to the internet. Based on this scenario, carrying out academic activities in a virtual way was a challenge for both students and professors.

This change in the teaching process forced Brazilian students and professors into a behavioral change, whereby the vast majority had to adapt their lives and spaces in order to prevent infection. However, in this new, challenging environment various problems arose. On the part of students, the reality of a lack of self-discipline, adequate learning materials or good learning environments when isolated at home was faced [42]; on the professors' side, adaptation to the overload of assigned work, student dissatisfaction, limited (or non-existent) students' access to necessary technologies, shortage of time for planning in digital media were encountered [43].

Freire [44] states that teaching and learning are distinct processes however interdependent and complementary. Additionally, this transitional educational process promoted by emergency remote teaching directly affects the community involved. Students and professors suffered from the overload imposed by this new dynamic, as individuals from different realities are part of these groups. People who carried out their activities at home

are mentioned. For them, it was possible to maintain isolation and adapt their routine to a domestic reality. Another group of individuals had to expose themselves to the virus to care for others or to provide essential services. Additionally, there were those who could not be isolated, as they needed to think about daily survival, even before viral contamination. Therefore, not everyone involved in this pandemic scenario had the same physical and material structure to manage the adversities being posed in their educational process. Thus, different student perceptions were to be expected regarding the quality of the teaching and learning process. In view of the above, the research methodology is presented below, with detailed data collection and analysis.

## 3. Research Methodology

In order to achieve the objective of this research work, which was to evaluate the ecological footprint and the perception of students about the teaching quality of online classes during the period of the SARS-CoV-2 pandemic, exploratory research was conducted at UFPA, which is the largest educational institution in the Brazilian Amazon.

### *Data Collection*

Data were collected between 13 March and 15 October 2021, whereby aspects associated with the period before the pandemic were analyzed based on the perception of the respondents. Students of undergraduate and graduate courses in civil and sanitary engineering at UFPA were the targets of this study. This research was approved by the Research Ethics Committee (REC) of the institution, according to the consolidated opinion number 4,459,236, issued on 12 December 2020.

A survey was performed with the students of the courses targeted by the research in order to understand the habits of the respondents before and during the pandemic period, as well as information on the perception of the learning process in the pandemic period. This survey was carried out by sending electronic forms to the students and consisted of 3 sequential steps which are detailed below:

1. The first stage aimed to collect information about the student's profile, perception of the learning process and performance during online classes, and whether the activities offered by the program were synchronous and/or asynchronous;
2. In the second stage, students were invited to answer a questionnaire about their habits before the COVID-19 pandemic, i.e., during the period of face-to-face classes. The questions were adapted from the Global Footprint Network® (Oakland, CA, USA) tool;
3. In the third stage, the respondents were required to respond to the same questionnaire as in stage II; however, they were asked to provide information regarding their habits during the COVID-19 pandemic and the resultant period of virtual classes;
4. In the fourth stage of this research, an analysis was carried out using the generalized estimating equations (GEE), where we aimed to identify whether there was a significant difference between the resources consumed before and during the COVID-19 pandemic through the measurement of the ecological footprint.

The data obtained were used to calculate the ecological footprint of each student before and during the pandemic using the Global Footprint Network® tool. Once the students answered the questionnaires, the data were assessed in a descriptive way and through inferential statistics by using the technique of the generalized estimating equation (GEE) to better understand and support the research result.

After establishing the methodology, a case study was developed and is explained in the following Section.

## 4. Case Description

The case area in this study is located in the city of Belém, in the Brazilian Amazon (see Figure 1). The case study took place in the largest higher education institution in the Eastern Amazon and covered the target audience of Civil Engineering and Sanitary and

Environmental Engineering courses. These two groups of students were chosen because of the similarity between their basic curricular structures. This choice was made due to specific issues, such as the collaboration of both the coordinator and the students of the course to participate in the research. Table 2 shows the profiles of respondents regarding the course, gender, age group and resources used to attend the virtual classes. In this sense, a questionnaire was sent to the students via Google Forms (see Supplementary Materials) that sought to gather information about the habits of the interviewees in the period before and during the COVID-19 pandemic.

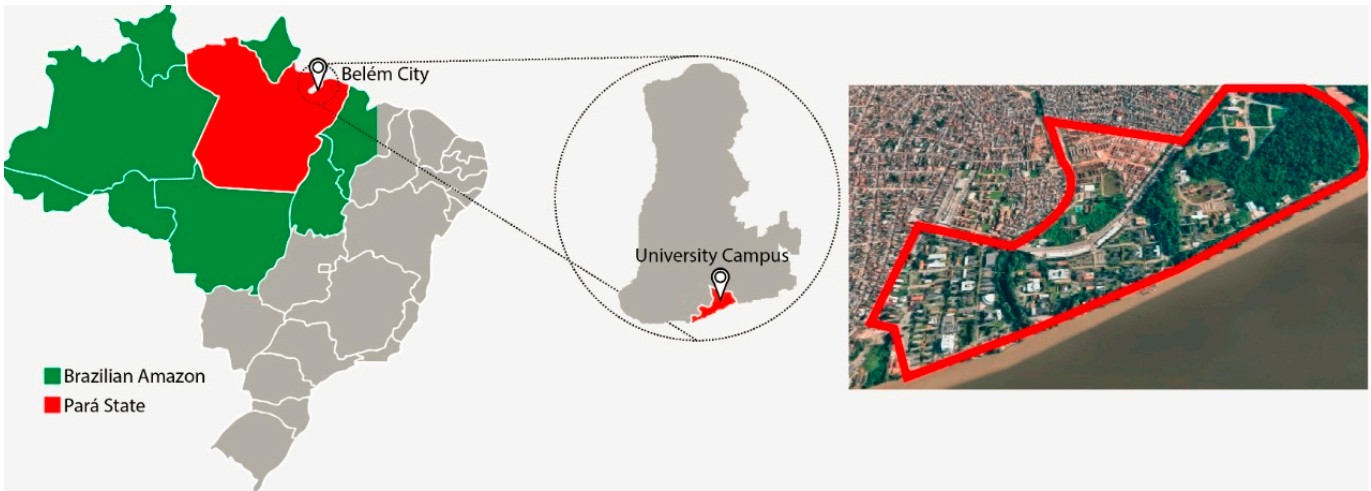

**Figure 1.** Case study location (Federal University of Pará-Brazil).

**Table 2.** Social demographic profile of respondents.

| Variable | | Frequency |
|---|---|---|
| Course | Civil Engineering | 117 |
| | Sanitary and Environmental Engineering | 32 |
| Sex | Male | 101 |
| | Female | 46 |
| | Non-binary | 2 |
| Age group | Up to 20 years | 23 |
| | Between 21 and 25 years old | 94 |
| | Between 26 and 30 years old | 13 |
| | Over 30 years old | 19 |
| Resources used to follow classes | Cell phone only | 51 |
| | Laptop only | 71 |
| | Computer only | 14 |
| | Computer and cell phone | 2 |
| | Laptop and cell phone | 8 |
| | Cell phone and tablet | 1 |
| | Computer, laptop and cell phone | 2 |

After this survey, the respondents' assessment was analyzed regarding the level and quality of their learning in the context of classes held virtually, as well as the tiredness felt during this period and the volume of tasks performed. Information was also collected on food, housing, mobility, and the consumption of goods and services before the pandemic in order to feed the Footprint calculation software and obtain the total ecological footprint. The results are presented and discussed in the following paragraphs.

## 5. Results and Findings

In the first stage of the research, to answer the first research question that analyzes the impact of the educational quality of online teaching it can be seen that among the results



found in the research in relation to the actions that should be implemented to improve online classes, the higher incidence of responses converge to actions such as (1) the need for professors to record classes so that students can review or attend classes (this occurred mainly due to the instability of internet connection); (2) professors reducing the amount of extra-curricular essays; (3) the possibility of improving professors' didactics regarding the use of online platforms; (4) greater interaction between professors and students; and (5) greater flexibility in the way of teaching and transference of the content of the disciplines. Students' responses converge to a better training of professors regarding this new didactic approach regarding online classes. To better understand this new context, the results of the research in relation to the teaching quality are broken down below.

### 5.1. Teaching and Learning through Online Classes

Respondents were asked about the quality of internet connection, the quality of the classes, their own learning, institutional support, the volume of tasks compared to face-to-face classes and the feeling of tiredness during this period.

About 20% of respondents claimed not to have computer resources available to keep up with the classes. However, about 35% only used their cell phones for this purpose, which points to a primary problem for the implementation of online classes in institutions such as UFPA. This lack of resources issue was also described by Huber and Helm [45] in Germany at the beginning of the pandemic. Regarding connection quality, 42.2% reported that it was regular, while another 37.6% indicated that the connection was good. A similar result was found by Zajdel et al. [46] in Poland, where the connection was rated as 3.35 on a 1-to-5 scale. Only 10% of respondents considered the connection quality very good, whereas around 8% considered it very bad. The "very bad" extreme did not have a significant percentage of responses. At the same time, it can be seen that most respondents who claimed to have a bad or regular connection only used their cell phones as a resource to follow the virtual classes.

Regarding the quality of classes, 67% of respondents believed that the quality was poor when compared to face-to-face classes. Moreover, 5% considered the teaching quality to be very poor and 25% declared that the quality was maintained between face-to-face and virtual classes. The percentage of students who considered teaching in virtual classes to be of better quality is less than 0.2%, which reflects that this was not the reality of the courses under analysis. Low quality in virtual learning was also a concern in developed countries. Rapanta et al. [47] emphasizes that universities should invest in the professional development of professors, so that they are able to use appropriate pedagogical methods with or without the use of online technologies.

Regarding quality, the individual learning assessment was also studied with the respondents, who deemed it to be 49% lower, 20% much lower and 21% of the same level, when comparing virtual classes to face-to-face classes. This characteristic was also addressed by Novitasari et al. [48], who stated that it is necessary for educational institutions to offer exceptional training for the use of innovative technologies by their employees.

Therefore, virtual classes during the COVID-19 pandemic undoubtedly had a negative impact on the teaching and learning process of students in the undergraduate courses that were analyzed. One of the hypotheses for this negative performance is the absence of infrastructure for internet access in student's houses, as most students do not receive cell phone signal where they live [49].

This deficiency in terms of connectivity is not only a reality in Brazil. According to the UNESCO report [50], less than 50% of the inhabitants of Africa, Latin America, the Caribbean, Asia and the Pacific do not have an Internet connection at home. Additionally, they suffer from a shortage of equipment for remote classes.

Furthermore, when analyzing the public that only used a cell phone as a resource, it was concluded that 74.5% classified the quality of classes as poor or very poor and about 57% also categorized learning as poor or very poor. The conclusion is that cell phones are

not an appropriate electronic gadget for virtual classes which can have an impact on the quality of teaching and the level of student learning.

As for the volume of tasks during virtual classes, 31% of respondents indicated that a significant increase was observed; 25% stated that some increase occurred; about 22% stated that there was no change when comparing the period before and during the pandemic; 20% indicated that there was little increase in the volume of tasks; and only 2% reported that the increase in the volume of tasks was very little. At the same time, 53% of respondents indicated that their feeling of tiredness increased during the pandemic, 20% indicated that it remained the same and 26% stated that it was reduced during the pandemic. Based on these answers, it is possible to say that those involved in the study did not have a hybrid curriculum that would guarantee a successful process while exploring virtual learning environments. This situation should be taken as an opportunity for higher education institutions to prepare for the digital world.

According to Shahzad et al. [51], students' perception of the workload in the virtual period is that it increased, and this has contributed to dissatisfaction with this teaching model, generating anxieties and dissatisfaction. This leads one to wonder about the preparation of university professors to provide for an audience that suffers from learning anxiety. Sharma and Sarkar [52] commented that in crisis situations similar to this one, encouragement is needed to improve self-confidence and thus improve the teaching–learning process, but often the professors themselves are not provided with this type of preparation during their training process.

The research instrument that was used also questioned what type of teaching should be used if students could choose—65% of them indicated that they preferred face-to-face teaching, while 27% indicated that hybrid teaching (face-to-face and online) would be the best option. This dissatisfaction was to be expected, since the professors were exercising a teaching model for which they were not prepared. Rasheed et al. [53], stated that many professors struggled to teach due to technological illiteracy. Watermeyer et al. [54] commented on how challenging this period of remote work was for the professors who had to adapt their home into a workplace. They had to manage a greater workload and also balance their personal and emotional life. Additionally, the students also suffered because they had to adjust their academic commitments to the routine at home, often having to attend class without any privacy. Lassoed et al. [55] 2020 found that, despite discomfort, privacy was not a concern for the students and professors that were interviewed.

A similar result was also found in a research work at the University of Cádiz, whereby 65.4% of students believed that the professors were not adequately leading online classes [48]. Institutional support during virtual classes was also questioned as 64% of respondents rated it as regular, 20% as good, around 13% as bad and approximately 3% as very bad.

An inferential analysis was performed among the research variables in order to test the hypotheses in a descriptive way. It was found that there was a moderate (correlation coefficient of 0.464 with $p > 0.01$) and positive correlation between Class Quality and Learning, i.e., there were statistical indications that a relationship between the quality of classes and learning exists (if one increases, the other also increases), similar to results also found in the research by Zajdel et al. [46].

In the next topic, the results obtained for the ecological footprint before and during the COVID-19 pandemic are presented–with regard to face-to-face and virtual classes, respectively.

### 5.2. Ecological Footprint before and during the COVID-19 Pandemic

In this Section, the result of the quantification of the students' ecological footprint is demonstrated before and during the pandemic period, with the aim of clarifying the 2nd and 3rd research questions.

In this fourth stage, the result of the statistical analysis of the footprint is demonstrated for the scenario that preceded, and during the period of the pandemic for the students targeted by the research. The collected data originated from measurements before and during the period of social isolation of 149 individuals. The box plot (Figure 2) was used to

compare the distribution of the data with the result of the ecological footprint (measured in GHA) before the pandemic and during the pandemic.

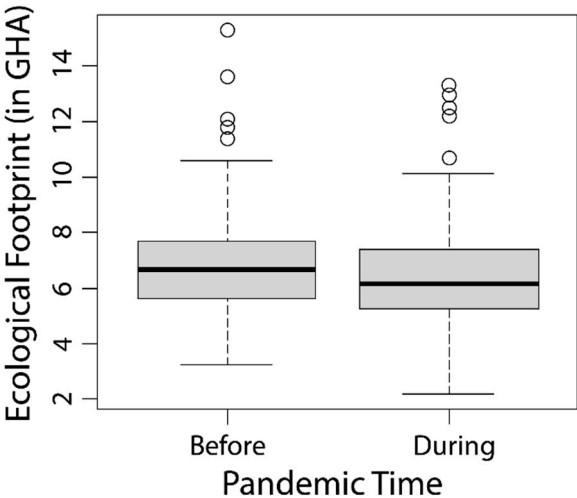

**Figure 2.** Box plot for the ecological footprint (in GHA) in the period before and during the COVID-19 pandemic.

Through the box plot chart, it can be seen that there was a reduction in the ecological footprint during the implementation of virtual classes due to the pandemic compared to the period of face-to-face classes before the pandemic. However, it was imperative to use an advanced statistical technique due to the small variability of the results found, in order to obtain greater relevance for the analysis of the collected information. For this, the GEE model was used to assess whether differences exist between the two periods (see Table 3).

**Table 3.** GEE model for ecological footprint analysis (in GHA) before and during the COVID-19 pandemic.

| Parameters | Estimation | Standard Error | Wald Test | $p$ * |
|---|---|---|---|---|
| intercept | 7.2453 | 0.3408 | 452,075 | $<2 \times 10^{-16}$ |
| Time | −0.3872 | 0.2130 | 3303 | 0.0691 |

* $p$-value.

The results of the adjusted model showed that there was no statistically significant difference between the times before and during the pandemic, i.e., $p < 0.05$. A similar result also occurred in the study carried out in Indonesia on the reduction in tourists on the islands; despite the reduction in visitors, there was no significant reduction in $CO_2$ emission in the region [56].

Even though the result of the ecological footprint calculation did not show a significant difference statistically, it is worth mentioning that nowadays any and all reductions are important to contribute to the mitigation of environmental impacts. Therefore, the results were explored by using the indicators that contribute to the footprint, in order to verify which components underwent greater variation between the two scenarios. In this sense, a new analysis was performed using the box plot chart (Figure 3) and the GEE model (Table 4) containing the variables by category of consumption (food, housing, mobility, goods and services), and again comparing the periods before and during the pandemic.

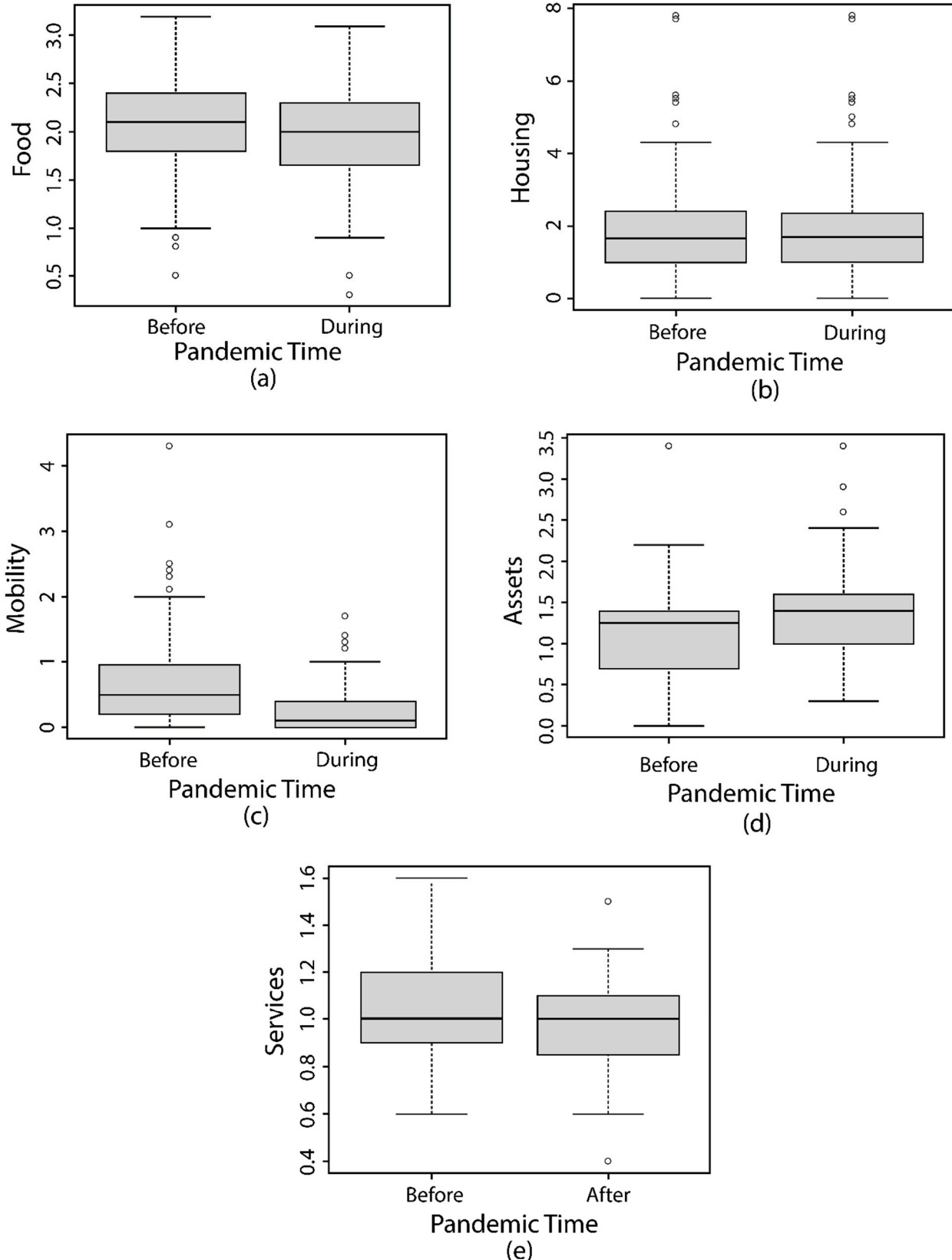

**Figure 3.** Box-plot of variables by consumption category before and during the pandemic: (**a**) food; (**b**) housing; (**c**) mobility; (**d**) goods; and (**e**) services.

**Table 4.** Analysis of differences in consumption by categories for the periods before and during the COVID-19 pandemic (Generalized Estimating Equations).

| | | **Food** | | |
|---|---|---|---|---|
| Parameters | Estimation | Standard Error | Wald test | *p* * |
| intercept | 2.1858 | 0.0892 | 600.59 | $<2 \times 10-16$ |
| Time | −0.0939 | 0.0584 | 2.59 | 0.11 |

| | | **Housing** | | |
|---|---|---|---|---|
| Parameters | Estimation | Standard Error | Wald test | *p* * |
| intercept | 1.9095 | 0.2568 | 55.3 | $1.00 \times 10-13$ |
| Time | −0.0101 | 0.1628 | 0 | 0.95 |

| | | **Mobility** | | |
|---|---|---|---|---|
| Parameters | Estimation | Standard Error | Wald test | *p* * |
| intercept | 1.105 | 0.117 | 89.7 | $<2.00 \times 10-16$ |
| Time | −0.419 | 0.064 | 42.8 | 6.10E-11 |

| | | **Goods** | | |
|---|---|---|---|---|
| Parameters | Estimation | Standard Error | Wald test | *p* * |
| intercept | 0.927 | 0.0873 | 112.8 | $<2.00 \times 10-16$ |
| Time | 0.2054 | 0.0594 | 11.9 | 0.00055 |

| | | **Services** | | |
|---|---|---|---|---|
| Parameters | Estimation | Standard Error | Wald test | *p* * |
| intercept | 1.1345 | 0.0372 | 930 | $<2.00 \times 10-16$ |
| Time | −0.0777 | 0.0224 | 12 | 0.00053 |

* *p*-value.

It was observed that in this case there were statistically significant differences, with a significance level of 5%, in the following variables: mobility, goods and services. For the others, the model showed no statistically significant difference, i.e., *p* > 0.05, as can be seen in Table 4.

In parallel, a direct numerical analysis of the difference between the ecological footprint before and during the pandemic was also developed, which can be seen in Table 5. The results lead to the conclusion that food and housing are the most significant patterns in the calculation of the total ecological footprint. At the same time, mobility is the pattern with the biggest difference in ecological footprint (61% reduction)—which can be explained by the drastic change in student behavior and mobility patterns during the pandemic. This proves the importance of this matter, as the transport of materials has a relevant impact on the region in terms of greenhouse gas emissions [57]. In this scenario, as the classes became virtual, a large part of the students completely reduced their daily travels. It can also be seen that the ecological footprint was reduced by 2.33% during the pandemic, a reduction that can be attributed mainly to mobility, but which reduced in proportion compared to the negative impacts generated in education and presented in item Section 5.1.

**Table 5.** Ecological footprint before and during the COVID-19 pandemic.

| Groups | Previous Ecological Footprint (GHA) | Ecological Footprint during the Pandemic (GHA) | Variation |
|---|---|---|---|
| Food | 2.26 | 2.29 | +1.3% |
| Housing | 2.02 | 2.07 | −2.47% |
| Mobility | 0.95 | 0.37 | −61% |
| Goods | 1.31 | 1.67 | +27.5% |
| Services | 1.18 | 1.14 | −3.4% |
| Total (GHA) | 7.72 | 7.54 | −2.33% |

The analyses developed made it possible to infer that the differences in the ecological footprint before and during the pandemic in this sample presented reduced values, despite having managed to conclude that there was a statistical difference between them. These results motivate the discussion about the reduction in environmental impacts and dimensions considered favorable for this reduction. Considering that any reduction—however small—is important, the results can be considered relevant.

Although this research was carried out in only one university, which is the largest university in the Brazilian Amazon, it is worth noting that the objective was not to exhaust the subject, but to contemplate the characteristic reality of the region, as well as other research works such as that of Zajdel et al. [46] and Rapanta et al. [47] who analyzed similar issues in only the universities in which they worked.

However, it can be demonstrated that in this case in the Amazon, it was evident that according to the perception of the students who were the target of the research, there was a significant loss in the quality of learning, whether in relation to the need for greater training on the part of the professors, the greater availability of resources for providing classes, among other characteristics previously demonstrated. In addition, the preference of 65% of interviewees for conducting face-to-face classes was evident; this fact demonstrates that learning did not achieve the desired performance with the change of classes remotely. On the other hand, despite the gain in mitigating the ecological footprint having obtained values with a more expressive degree of significance in terms of mobility and consumption of goods, the values achieved are lacking if compared to a lack of a significant loss in learning. The reduction in the footprint occurred only in a more expressive way with regard to the reduction in $CO_2$ emissions in relation to the transport of students from their homes to the educational institution (see Table 5), can be corroborated by the recent research published by the authors Potenze et al. [58], of the climate observatory, who claimed that the energy sector had a strong reduction in $CO_2$ emissions (4.6%), especially due to the recession and social isolation in the first semester, which brought down the consumption of gasoline in the transport of passengers. However, on the other hand, consumption of goods increased (see Table 5) because, due to the fact that students remained at home, the consumption of food, energy and the acquisition of electronic equipment to develop activities related to study and leisure remotely increased. This result found in the research was also corroborated by the authors Potenze et al. [58], as their research stated that the increase in greenhouse gases in the industrial processes and use of products remained practically stable, with an oscillation of 0.5% upwards, as opposed to energy.

## 6. Conclusions and Future Works

It is undeniable that the pandemic process had consequences in all sectors. Evaluating this impact on the educational process is relevant for all educational institutions in the world. However, the pedagogical process when teaching classes remotely, in addition to changing the routine and the quality of teaching, provided a reduction in the ecological footprint, a fact that was made evident in this study in the Amazon region.

The methodology proposed has two main axes. The first aimed to collect the students' opinion on the quality of teaching and their difficulties regarding remote classes. The second axis aimed to quantify the students' ecological footprint with the change of routine before and during the pandemic period through an internationally used, validated tool (Global Footprint Network®). After a Survey of these two axes, a statistical treatment was performed for better understanding of the result achieved.

According to the students' opinion, a significant loss in the quality of virtual classes was observed when compared to face-to-face classes and, as a consequence, there was a reduction in relation to learning the content being taught. This result clearly showed that despite the institution's effort to provide the necessary resources for the online teaching modality to be widely used by students, numerous difficulties were reported, mainly problems with the quality of the internet, lack of more adequate methodologies during the

process of teaching and a greater tiredness on the part of the students with the asynchronous and synchronous tasks during this period.

On the other hand, the results indicated that the reduction in the ecological footprint was not significant in this period. When the types of consumption were assessed, the mobility item was highlighted with the greatest reduction among the other items, but in spite of that, it was not possible to find a reduction in the entire process. These values were calculated by using a methodology consolidated worldwide, where the tool had already been used in several papers published in high impact journals, a fact that corroborates the assertiveness of the tool used in the calculations of this research.

Although a reduction in environmental impacts was found, the negative aspects related to student learning were preponderant as a result of the pandemic scenario. There is also a need for the Amazonian institution to develop qualification programs for its professors to use virtual tools in the educational process, since, apparently, this may be a trend in the future and its professors need to adapt to this new work dynamic. As also shown in the research by Cioca and Bratu [59], the pedagogical performance of students reduced during online classes, and clear criticisms regarding the performance of professors when teaching online classes emerged, thus demonstrating the importance of implementing educational policies that can minimize these problems in the future.

**Supplementary Materials:** The following supporting information can be downloaded at: https://www.mdpi.com/article/10.3390/su14169891/s1, Questionnaire.

**Author Contributions:** Conceptualization, L.M.F.M., F.d.S.M., R.L.d.P.S.; methodology, L.M.F.M., R.L.d.P.S., L.d.N.P.C.; software, R.L.d.P.S., F.d.S.M., P.C.d.S.J.; validation, L.M.F.M., R.L.d.P.S., L.d.N.P.C., O.M.P.P.; formal analysis, L.M.F.M., F.d.S.M., L.d.N.P.C.; resources, L.M.F.M., F.d.S.M., L.d.N.P.C.; writing—L.M.F.M., F.d.S.M., L.d.N.P.C., O.M.P.P.; writing—review and editing, F.d.S.M., L.d.N.P.C., O.M.P.P., R.L.d.P.S., P.C.d.S.J.; visualization, F.d.S.M., O.M.P.P.; supervision, L.M.F.M.; project administration, L.M.F.M.; funding acquisition, L.M.F.M. All authors have read and agreed to the published version of the manuscript.

**Funding:** The authors received no financial support for the research, but received financial support for the publication of this article from PROPESP/PAPQ/UFPA. This study was financed in part by the Coordenação de Aperfeiçoamento de Pessoal de Nível (CAPES) (Coordination for the Improvement of Higher Education Personnel)-Brazil-Finance Code 001.

**Institutional Review Board Statement:** The study was conducted in accordance with the Declaration of Helsinki, and approved by the Institutional Review Board (or Ethics Committee) of Federal University of Pará (protocol code 4.459.236–12/12/2020).

**Informed Consent Statement:** Not applicable.

**Data Availability Statement:** Not applicable.

**Conflicts of Interest:** The author declares that there are no potential conflicts of interest with respect to the research, authorship, and/or publication of this article.

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
