# Peer review of "Impact on Education and Ecological Footprint as a Consequence of SARS-CoV-2 in the Perception of the Quality of Teaching Engineering Students in the Brazilian Amazon"

_sustainability, doi:10.3390/su14169891_

Round 1
Reviewer 1 Report
It is a very ambitious research plan and a good topic, but it would be important to define the boundaries of the research more clearly.
Why is it interesting that engineering students were studied? Would the results be different for other groups of students? To what extent is this group of students different or similar to others? (It would be good to find literature sources on this.) What was the exact questionnaire? What was included in the consumption? (For example, did they take into account that universities that close do not need heating/lighting and that although home/rental increases students' housing costs and consumption, but decreases community consumption.)
In my opinion, there are a number of questions that remain open that can be avoided by explaining the data collection accurately.
It could be valuable and significant work, but there is still much to be done in terms of delineating and documenting the research.
Author Response
Thank you for your suggestions for improving the article, we try to answer your questions. I hope we made it.

Reviewer 2 Report
1. The study did not adequately address the scientific need to examine the linkage between educational approaches and ecological footprints in the context of the epidemic, e.g., what is the potential value of exploring this topic and the under-representation of case studies through only one university and limited majors.
2. The literature review failed to support the relevant background related to education-based ecological footprint analysis.
3. The methodology is lack of the information on ecological footprint analysis, which should involve essential data source, index system, algorithm, etc.
4. This paper said the data was collected during March-October, 2021, how could they get the information before the pandemic?
5. More arguments should be provided in this paper to support their findings and potential implications for current collage education, rather than presenting their localized results only.
6. Is there any figures to support the spatial location of your case area? And this study should provide more detailed information of their interview.
Author Response

(The authors gave the same response as above.)

Reviewer 3 Report
Highlight changes in yellow in a next revision, please. No track changes.
Please address the country here as in the title?
“and environmental impacts of the academic community of the largest educational institution in the 18
Amazon region.”
Consider using excessed instead of evaluated.
Please clarify the findings and practical implications. Also addressing innovation and novelty.
“In environmental 24
terms, the reduction of the ecological footprint was insignificant, despite a reduction in a specific 25
sector such as mobility.”
Please carefully revised reference format.
“Wuhan (1).”
Please try to better connect the content instead of including such separated paragraphs.
“This chaotic scenario lived in society also generated restrictions in the educational 51
area due to the prohibition of face-to-face classes through decrees. Educational institu- 52
tions, seeking to minimize the impact generated by the interruption of face-to-face classes, 53
sought alternatives with the adoption of virtual classes. This sudden change in pedagog- 54
ical processes forced educators and students to undergo adaptations in the educational 55
routine. 56
In a survey carried out by McFadden et al. (8), who sought to assess the impact of the 57
pandemic on universities in Australia, England, Finland, Northern Ireland, Norway, Ire- 58
land and Sweden, there are changes both in the admission phase of these institutions and 59
in the implementation of virtual classes for students of the social sciences course. This 60
change in behavior was recurrent both in universities in developed and developing coun- 61
tries.”
All over the manuscript.
Please carefully address spacing and subscripts as italics later in parameters.
“(CO2 ),”
Captions must be self-explanatory by their own, and this is not. Also, it would be important to add a link and URL to each software being addressed.
“Table 1. Software for calculating the ecological footprint.”
Are these results then? Unclear.
“4. Case description”
Captions are not clear enough. Do you refer to social demographic profile? please better clarify.
“Table 2. Profile of respondents.”
This is not that easy as I would avoid this kind of listing.
“In the next topic, the results obtained for the ecological footprint before and during 343
the COVID-19 pandemic will be presented – with face-to-face and virtual classes, respec- 344
tively. These will be analyzed in light of the impacts presented in this topic regarding the 345
teaching quality.”
But see that this is methodology.
“5.2 Ecological footprint before and during the COVID-19 Pandemic 347
In the second stage of this research, an analysis has been initially carried out using 348
the generalized estimating equations (GEE), where was aimed to identify if there is a sig- 349
nificant difference between the resources consumed before and during the COVID-19 350
pandemic, through the measurement of the ecological footprint.”
Is my perspective that this figure, who has low quality needs to be improved both in terms of quality as in terms of content. For example time meaning exactly what.
“Figure 1: Box plot for the ecological footprint (in GHA) before and during the pandemic. 2
Lower p and dash
“P value”
Please check all italics.
“p < 0.05.”
The figure has very low quality. See that in case of group figures. A different letter must correspond to each one, and then a subcaption must exist after the main caption.
“Figure 2: Box-plot of variables by consumption category before and during the pandemic”
The caption needs to better translate to the content of the table.
“Table 4. GEE model for analysis of consumption categories before and during the pandemic.”
Please start with a brief contextualization as well as a brief methodology. Does defending your manuscript.
“6. Conclusions and future works 407
The consequences of the pandemic process caused by SARS-CoV-2 with regard to the 408
impact on the educational process quality and students’ learning in the largest public 409
higher education institution in the Amazon, as well as the impact on the students’ ecolog- 410
ical footprint, were the topics explored in this study.”
Unclear.
“Despite the finding of small benefits related to the reduction of environmental im- 427
pacts, the negative aspects related to student learning were preponderant as a result of 428
the pandemic scenario. ”
The sentence cannot start like this.
“As well as the need”
The manuscript will need entire English revision.
The language and content is unclear, read again
“Finally, as future works, the development of studies that can try to relate the impact 433
of this learning gap, which occurred in the formation of this institution's students, is rec- 434
ommended and its application after the exercise of the professional function.”
Despite the regional interest of this manuscript, the authors need to better translate it into a relevant.
It would be important to translate the findings to be reflected at a wider scope.
It would be important to add numbers to the “research questions:” in order to be more specifically addressed in the results/discussion and conclusions, as in the abstract.
Author Response

(The authors gave the same response as above.)

Round 2
Reviewer 1 Report
The article is significantly clearer and more concise as a result of the changes.
Author Response
Thank you very much for your suggestions in improving our article.
Reviewer 2 Report
I cannot be convinced by the current revisions of this paper, most of the key information that I mentioned in last review are still lacking.
Author Response
Dear reviewer,
We've made several adjustments to the text to improve its content. As their remarks were not specific, it became more difficult to try to answer them. We did the grammar review in English and changed the content in the text in a broader way in an attempt to improve our research and in this way we could contemplate your suggestions.
In this sense, we base ourselves on other articles that have been published in the sustainability journal. We hope we could have met your demands.
Reviewer 3 Report
Highlight changes in yellow in a next revision, please. No track changes.
I do not understand what happened since my comments were extensive and the cover letter from the authors stopped here:
“Please try to better connect the content instead of including such separated paragraphs.
“This chaotic scenario lived in society also generated restrictions in the educational area due to the prohibition of face-to-face classes through decrees. Educational institutions, seeking to minimize the impact generated by the interruption of face-to-face classes, sought alternatives with the adoption of virtual classes. This sudden change in pedagogical processes forced educators and students to undergo adaptations in the educational routine. In a survey carried out by McFadden et al. (8), who sought to assess the impact of the
pandemic on universities in Australia, England, Finland, Northern Ireland, Norway, Ire- land and Sweden, there are changes both in the admission phase of these institutions and in the implementation of virtual classes for students of the social sciences course. This change in behavior was recurrent both in universities in developed and developing countries.”
All over the manuscript.
”
The ones below had no answer and need to be addressed.
They are available in the site, I just checked
Please carefully address spacing and subscripts as italics later in parameters.
“(CO2 ),”
Captions must be self-explanatory by their own, and this is not. Also, it would be important to add a link and URL to each software being addressed.
“Table 1. Software for calculating the ecological footprint.”
Are these results then? Unclear.
“4. Case description”
Captions are not clear enough. Do you refer to social demographic profile? please better clarify.
“Table 2. Profile of respondents.”
This is not that easy as I would avoid this kind of listing.
“In the next topic, the results obtained for the ecological footprint before and during 343
the COVID-19 pandemic will be presented – with face-to-face and virtual classes, respec- 344
tively. These will be analyzed in light of the impacts presented in this topic regarding the 345
teaching quality.”
But see that this is methodology.
“5.2 Ecological footprint before and during the COVID-19 Pandemic 347
In the second stage of this research, an analysis has been initially carried out using 348
the generalized estimating equations (GEE), where was aimed to identify if there is a sig- 349
nificant difference between the resources consumed before and during the COVID-19 350
pandemic, through the measurement of the ecological footprint.”
Is my perspective that this figure, who has low quality needs to be improved both in terms of quality as in terms of content. For example time meaning exactly what.
“Figure 1: Box plot for the ecological footprint (in GHA) before and during the pandemic. 2
Lower p and dash
“P value”
Please check all italics.
“p < 0.05.”
The figure has very low quality. See that in case of group figures. A different letter must correspond to each one, and then a subcaption must exist after the main caption.
“Figure 2: Box-plot of variables by consumption category before and during the pandemic”
The caption needs to better translate to the content of the table.
“Table 4. GEE model for analysis of consumption categories before and during the pandemic.”
Please start with a brief contextualization as well as a brief methodology. Does defending your manuscript.
“6. Conclusions and future works 407
The consequences of the pandemic process caused by SARS-CoV-2 with regard to the 408
impact on the educational process quality and students’ learning in the largest public 409
higher education institution in the Amazon, as well as the impact on the students’ ecolog- 410
ical footprint, were the topics explored in this study.”
Unclear.
“Despite the finding of small benefits related to the reduction of environmental im- 427
pacts, the negative aspects related to student learning were preponderant as a result of 428
the pandemic scenario. ”
The sentence cannot start like this.
“As well as the need”
The manuscript will need entire English revision.
The language and content is unclear, read again
“Finally, as future works, the development of studies that can try to relate the impact 433
of this learning gap, which occurred in the formation of this institution's students, is rec- 434
ommended and its application after the exercise of the professional function.”
Despite the regional interest of this manuscript, the authors need to better translate it into a relevant.
It would be important to translate the findings to be reflected at a wider scope.
It would be important to add numbers to the “research questions:” in order to be more specifically addressed in the results/discussion and conclusions, as in the abstract.
Author Response
Dear reviewer, we are submitting the adjustments we made to the text, we apologize on the occasion of the first round as I do not know what happened, but it did not appear for us when we downloaded all your observations. We try to do our best to serve you at this time, we hope we have succeeded.

Round 3
Reviewer 2 Report
They have done dense of revisions
Author Response
Dear reviewer,
We've made more tweaks to the text to improve your content. As well as the revision of the text in English was redone once again. We hope that the text has improved to meet your expectations with its content.
Reviewer 3 Report
Highlight changes in yellow in a next revision, please. No track changes.
Please do not track change the document. Even if not using the tool and just cutting, this is not the way to do it. It becomes almost impossible to check general coherence.
Example
“
|
The impact that the severe acute respiratory syndrome (SARS-CoV-2) responsible for |
32 |
|
the coronavirus disease (COVID-19) has had on the planet Earth since its spread it started |
”
In the perception of exactly what. Read the title again.
“in the perception of engineering stu- 3
dents”
I am not asking of who, but of what…
Read again…
“The perception of undergraduate 19
and graduate students in the period of during remote teaching was assessed using by means of an 20
exploratory research carried out at the Federal University of Pará.”
The perception towards what?
Check other cases.
See that the country should be added to University of Para.
You do not mention to be engineering students here.
“The perception of undergraduate 19
and graduate students in the period of during remote teaching was assessed using by means of an 20
exploratory research carried out at the Federal University of Pará.”
More expressive than which ones?
“the reduction of the ecological footprint that occurred was observed could 26
have reached more expressive values,”
This is the first time I clearly understand too which perception are the authors refering to.
Assure to be clear through the entire text.
“
|
1- What is the student's perception regarding the quality of the classes taught and the |
134 |
|
level of learning with the adoption of by adopting virtual classes? |
”
Be more clear.
“
|
Thus, different perceptions |
258 |
|
regarding their learning are to be expected. |
”
Figure have captions below and tables above. Done like this almost everywhere…
Do not add abbreviations to a caption. That forced the reader to go and find information within the text.
“Figure 1. Case study location (UFPA)”
The expression pandemic time means exactly what_ See that this article will be read 20 years from now and readers will not understand to which time period are you refering to.
In fact, an, unless I am mistaken, we are still living in the pandemic. So be clear every time.
“Figure 1: Box plot for the ecological footprint (in GHA) in the period before and during the pan- 452
demic time”
So unless I am completely wrong, figure two is completely distorted. Just check it please.
“Figure 2: Box-plot of variables by consumption category before and during the pandemic: (a) food; 478
(b) housing; (c) mobility; (d) goods and (e) services.”
In table 4, “p” should appear in the table and then in notes you would define that p means be p/value. Done like this everywhere.
If this question cannot be verified, it should not be here, should it?
“It should be noted that the fourth research question, "Is there a relationship between 579
learning and reducing the size of the ecological footprint?", cannot be verified in this re- 580
search. Therefore, for so as future works, it is recommended the development of develop- 581
ing studies that can try to report these questions is recommended.”
Think about this because it is important.
Please read your conclusions again. See that after the contextualization in yellow, you immediately start discussing the results without mentioning the methodology in a clear way. There is only a mention to the Global Footprint network.
“6. Conclusions and future works 537
It is undeniable that the pandemic process had consequences for in all sectors is un- 538
deniable. Evaluating this impact on the educational process is relevant for all educational 539
institutions in the world. However, the pedagogical process when teaching classes re- 540
motely, in addition to changing the routine and the quality of teaching, provided a reduc- 541
tion in the ecological footprint, a fact that was made evident in the herein study in the 542
Amazon region through the realization of a Survey and the by using the calculator pro- 543
vided by the Global Footprint Network®. 544
The consequences of the pandemic process caused by SARS-CoV-2 with regard to the 545
impact on the educational process quality and students’ learning in the largest public 546
higher education institution in the Amazon, as well as the impact on the students’ ecolog- 547
ical footprint, were the topics explored in this study. It has been identified, According to 548
the students’ opinion, it was observed that there was a significant loss in the quality of 549
virtual classes when compared to face-to-face classes and, as a consequence, there was a 550
reduction in relation to the learning of the taught content being taught.”
This is just not written like this. It can be seen that the authors are not very. Used to the English scientific language.
“Despite the finding of small reduction”
See that my comments. Only intend to contribute to a relevant paper. Everything you do or live in a paper is significant or should be...
I would advise the authors to read into a text once again and try to improve the language and coherence.
Language does need further investment…
See that authors should avoid using a language that will not be understood by most readers. Even in these cases. Surely it can be translated.
“
|
Acknowledgments: This study was financed in part by the Coordenação de Aperfeiçoamento de |
597 |
|
Pessoal de Nível Superior - Brasil (CAPES) - Finance Code 001. |
”
Each time the reference is in another language different from the one the article is being written, then the translation should be added inside square brackets. Added to the original language, not replace it.
“IESALC, UNESCO. Cerrar ahora para reabrir mejor mañana?. Perfiles Educati-vos”
So that the reader may understand it.
There are more cases.
What is this_
“Bao W. <scp>COVID</scp> -19”
Author Response
Dear reviewer
We are once again sending the article for your evaluation, we try to answer your questions, which certainly help to improve the quality of the article. We hope that this time it can be positively evaluated with a view to eventual publication.
Thanks for your attention.
